# Decellularized Pancreatic Tail as Matrix for Pancreatic Islet Transplantation into the Greater Omentum in Rats

**DOI:** 10.3390/jfb13040171

**Published:** 2022-09-30

**Authors:** Zuzana Berkova, Klara Zacharovova, Alzbeta Patikova, Ivan Leontovyc, Zuzana Hladikova, David Cerveny, Eva Tihlarikova, Vilem Nedela, Peter Girman, Daniel Jirak, Frantisek Saudek

**Affiliations:** 1Laboratory of Pancreatic Islets, Institute for Clinical and Experimental Medicine, 14021 Prague, Czech Republic; 2Diabetes Center, Institute for Clinical and Experimental Medicine, 14021 Prague, Czech Republic; 3First Faculty of Medicine, Charles University, 12108 Prague, Czech Republic; 4Radiodiagnostic and Interventional Radiology Department, Institute for Clinical and Experimental Medicine, 14021 Prague, Czech Republic; 5Faculty of Health Studies, Technical University of Liberec, 46117 Liberec, Czech Republic; 6Environmental Electron Microscopy Group, Institute of Scientific Instruments, The Czech Academy of Sciences, 61264 Brno, Czech Republic

**Keywords:** pancreas decellularization, splenic vein perfusion, extracellular matrix skeletons, transplantation into the omentum, advanced environmental scanning electron microscopy

## Abstract

Infusing pancreatic islets into the portal vein currently represents the preferred approach for islet transplantation, despite considerable loss of islet mass almost immediately after implantation. Therefore, approaches that obviate direct intravascular placement are urgently needed. A promising candidate for extrahepatic placement is the omentum. We aimed to develop an extracellular matrix skeleton from the native pancreas that could provide a microenvironment for islet survival in an omental flap. To that end, we compared different decellularization approaches, including perfusion through the pancreatic duct, gastric artery, portal vein, and a novel method through the splenic vein. Decellularized skeletons were compared for size, residual DNA content, protein composition, histology, electron microscopy, and MR imaging after repopulation with isolated islets. Compared to the other approaches, pancreatic perfusion via the splenic vein provided smaller extracellular matrix skeletons, which facilitated transplantation into the omentum, without compromising other requirements, such as the complete depletion of cellular components and the preservation of pancreatic extracellular proteins. Repeated MR imaging of iron-oxide-labeled pancreatic islets showed that islets maintained their position in vivo for 49 days. Advanced environmental scanning electron microscopy demonstrated that islets remained integrated with the pancreatic skeleton. This novel approach represents a proof-of-concept for long-term transplantation experiments.

## 1. Introduction

An established treatment for highly selected patients with type 1 diabetes that have hypoglycemia unawareness syndrome [1] is to transplant pancreatic islets by injecting viable islets through branches of the portal vein into the liver. Nevertheless, the localization of islets in small capillaries in the liver is far from optimal. Islet survival is negatively affected by an instantaneous blood-mediated inflammatory reaction, low oxygen tension, and excessive concentrations of orally administrated immunosuppressive drugs. Consequently, current research has been focused on finding new sites [2] for islet transplantation, with or without the use of specific biocompatible devices [3,4] such as extracellular matrix (ECM) scaffolds to avoid direct contact with recipient blood.

The ECM is a three-dimensional network of highly organized fibrous proteins and polysaccharides that provide dynamic microenvironment for different cell populations [5]. Currently, it is known that the ECM is important for structural support and for its essential role in physiological and pathological cellular processes, such as cell growth, migration, proliferation, differentiation, and apoptosis [6,7]. Developments in this area have progressed from the use of purified ECM components as a coating for cell cultures to ECM scaffolds with more-or-less preserved 3D structures, obtained with decellularization via various physical, chemical, and enzymatic treatments [8,9].

Biomedical research in pancreas decellularization has been focused either on the production of pancreatic skeletons with intact spatial architecture and vasculature [10,11] comparable to native pancreatic structures, or on the production of hydrogels [12] that contain ECM components. Many different detergent solutions, alone or in combination with enzymes—which can be helpful in removing remnant nucleic acids after cell lysis [8,13]—have already been tested for successful pancreas decellularization [10,11,14,15]. The strongest detergent needed for removing nuclear and cellular components is an ionic sodium dodecyl sulphate (SDS) [8,9,13]. However, SDS residues remaining in the skeleton could be potentially dangerous to repopulated cells [9,16]. Therefore, an excessive washing is crucial [16] to remove SDS residues from pancreatic ECM skeletons and prevent cytotoxicity.

To date, there are two diverse approaches for repopulating pancreatic skeletons: infusing pancreatic cells into the ECM scaffolds through a pancreatic duct, vein, or artery [10,11,14,15,17,18], or reseeding pancreatic cells on an ECM scaffold surface [19,20]. To date, pancreatic skeletons have been reseeded with acinar cell lines [10,20], beta cell lines [10,14,15], endothelial cells [15,18], endothelial progenitor cells [17], stem cells [19], and as expected, with isolated Langerhans islets [11,14,18,19]. Successful cell engraftment within decellularized pancreas was achieved in all these studies.

The islet isolation process removes islets from their native environment, which disrupts mechanical support, vascular supply, and neural input, and damages islet–ECM interactions. It was suggested that the omental interstitial space could be a promising extrahepatic site for implanting isolated islets. This approach could provide a rich vascular supply but would avoid direct contact with the blood [21]. Experimental data [2,22,23] and first clinical experiences [24,25] have suggested that this approach has potential, but to date, it has not replaced hepatic placement. Therefore, we reasoned that pancreatic ECM skeletons, with accurate microstructure and biochemical composition, might provide an optimal tissue-specific environment for isolated islets. The present study aimed to prepare decellularized skeletons from native rat pancreases that could be repopulated with pancreatic islets, and subsequently, transplanted into a new, promising transplant site in the greater omentum. We evaluated different perfusion techniques according to the ability to preserve the 3D structure of the matrix, the structural protein composition, and the residual DNA content. Due to the fact that an entire pancreas skeleton occupies quite a large volume and that, currently, the omentum site is limited in size, we designed a new perfusion approach for decellularization that required only the pancreatic tail, and perfusion was performed via the splenic vein. Then, we compared this technique with previously described techniques. The second part of this study was focused on repopulating the ECM skeletons, and after transplantation into the omentum, assessing viability with multimodal imaging. The fate of the transplanted islets in ECM skeletons was followed with in vivo magnetic resonance imaging (MRI) and correlated with ex vivo results from histology and electron microscopy examinations.

## 2. Materials and Methods

### 2.1. Experimental Animals

Adult male Lewis rats (Charles River, Germany) were used as pancreas donors for the decellularization processes and for pancreatic islet isolation. Another set of adult male Lewis rats served as recipients for transplantation experiments. All experimental protocols were approved by the Experimental Animals Welfare Committee of the Institute for Clinical and Experimental Medicine and the Ministry of Health of the Czech Republic (approval no. 36/2018) in accordance with the Protection of Animals against Cruelty Act (no. 359/2012) of the Czech Republic, which corresponds to the European Parliament and Council directive 210/63/EU.

All experimental animals were housed and bred in conventional housing in the Experimental Facility of the Institute for Clinical and Experimental Medicine. Rats were maintained in approved cages (3–4 rats/cage) with free access to water and pelleted food. Stable temperature and humidity conditions were maintained, and the light regime was 12 h light:12 h dark. Donor pancreases were surgically excised under general intramuscular anesthesia (dexmedetomidine 0.09 mg/kg and ketamine 36 mg/kg body weight). Transplantation experiments were performed under general inhalation anesthesia (isoflurane 5%/2%), with an intramuscular application of butorphanol (1 mg/kg). After surgery, the pain was suppressed with subcutaneous application of analgesics (meloxicam, 1–2 mg/kg).

### 2.2. Pancreas Perfusion Techniques

Donor pancreases were perfused by cannulating the pancreatic duct from the duodenal or hepatic side, via a gastric artery and via a portal or splenic vein, with 24 G catheters (Becton Dickinson and Company, Franklin Lakes, NJ, USA). After a donor laparotomy, 1000 IU of heparin (Zentiva, Prague, Czech Republic) was injected into the inferior vena cava.

For perfusion via the pancreatic duct through the papilla (PDP; *n* = 16), a small incision was made in the duodenum, 0.5 cm below the papilla, and a catheter was inserted anterogradely into the common bile duct and fixed with a clamp. The opposite end of the duct was ligated to prevent leakage. Subsequently, the entire pancreas with the duodenum was carefully excised and removed from the abdominal cavity. Ex vivo, the duodenum was rinsed with physiological solution, and both ends were ligated.

For perfusion via the pancreatic duct from the hepatic side (PDH; *n* = 13), a catheter was inserted retrogradely into the common bile duct and fixed with a clamp. Subsequently, the entire pancreas with the duodenum was carefully excised and removed from the abdominal cavity. Ex vivo, the duodenum was rinsed with physiological solution, and both ends were ligated. The duct outlet into the duodenum was also closed with the clamp to prevent leakage.

For perfusion via the portal vein (PV; *n* = 19), a catheter was inserted retrogradely into the portal vein, sutured in place, and fixed with clamp. The celiac and gastric arteries and the splenic vessels were ligated, and the entire pancreas was carefully excised and removed from the abdominal cavity.

For perfusion via the gastric artery (GA; *n* = 12) [15], the left gastric artery was exposed, and a catheter was inserted, sutured in place, and fixed with clamp. The celiac, hepatic, and splenic arteries were also exposed. All vessels were ligated. Then, the pancreatic body with tail was filled with a detergent solution, excised, and removed from the abdominal cavity for the decellularization process.

For perfusion via the splenic vein (SV; *n* = 34), the pancreatic tail was exposed and excised, together with the spleen, lymph nodes, and omentum. In detail, the bowel was moved to the left side to expose the pancreatic tail, which was then turned until the distal side was facing up. A clamp was placed in the area of branching gastric and splenic veins. Then, the pancreatic tail was cut underneath the clamp to excise it and removed from the abdominal cavity. The splenic vein has several branches; for the decellularization process, we used the distal branch, along with 3–4 branches that entered the splenic hilus. Ex vivo, the SV was exposed, cannulated, and a catheter was sutured in place and fixed with a clamp. Then, the splenic branches and other vessels were ligated to prevent leakage. Subsequently, the spleen, omentum, and 3–4 small lymph nodes were removed from the pancreatic tail and the pancreatic tail was prepared for perfusion and decellularization.

All ligatures and sutures were performed with black braided silk, 7-0 (Mersilk™, Ethicon LLC, Raritan, NJ, USA).

### 2.3. Decellularization of the Pancreatic Tissue

Each cannulated pancreas was connected to the perfusion system. The perfusion system consisted of a peristaltic pump, tubes, and a chamber. The flow rate was set to 5 mL/min. The decellularization protocol included a 60 min perfusion with 1% Triton X-100, a 120 min perfusion with 0.5% sodium dodecyl sulphate, and then a 120 min perfusion with 1% Triton X-100 (both detergents from Glentham Life Sciences Ltd., Corsham, UK). Finally, the process was completed by perfusing with a DNase solution (0.4 U/L; Merck, Darmstadt, Germany) for 60 min. After decellularization, the extracellular matrix was washed with phosphate-buffered saline (PBS) supplemented with a 1% antibiotic, antimycotic solution for 18 h (both chemicals from Merck, Darmstadt, Germany).

### 2.4. Quality Test for the Decellularization Process

The quality of the decellularization process was compared with pancreases from healthy control rats and pancreases from diabetic rats. Diabetes was induced in overnight-fasted animals with an intraperitoneal injection of streptozotocin (60 mg/kg; Merck, Darmstadt, Germany) dissolved in cold 3.8% sodium citrate (pH 4.5).

DNA contents were quantified in decellularized tissues and in control pancreases from healthy and diabetic animals. Briefly, tissues were transferred into lysis buffer, homogenized with a sonicator, and the amount of total DNA was determined in triplicates with a NanoDrop™ spectrophotometer (Thermo Fischer Scientific, Waltham, MA, USA).

Insulin contents were measured in decellularized tissues and in control pancreases from healthy and diabetic animals. Briefly, tissues were frozen, then minced and homogenized in acid–ethanol. The hormone concentrations in extracts were measured in duplicates with the Rat Insulin Ultrasensitive ELISA kit, for ECM skeletons, and with the Rat Insulin ELISA kit, for pancreases (both from Mercordia, Uppsala, Sweden).

### 2.5. Cytocompatibility Testing of the ECM Skeletons

The ECM skeletons cytocompatibility was tested via dynamic cultivation of mesenchymal stem cells (MSCs) in the ECM skeletons (*n* = 3). The MSCs were isolated from retroperitoneal and epididymal fat from Lewis rats according to our standard protocol. Briefly, tissues were digested by collagenase solution (1 mg/mL; Merck, Darmstadt, Germany), and MSCs were then separated on Ficoll-Paque (1.077 g/mL; GE Healthcare Bio-Sciences AB, Uppsala, Sweden). Isolated MSCs were incubated in DMEM medium supplemented with 10% FBS, 5% HEPES, 1% penicillin–streptomycin, 1% GlutaMAX, 1% insulin–transferin–selenium (all chemicals from Merck, Darmstadt, Germany), and 50 ng/mL EGF (epidermal growth factor; R&D Systems, Minneapolis, MN, USA) in a humidified incubator at 37 °C and 5% CO_2_ atmosphere until passage 6. The MSCs were analyzed by flow cytometry (BD LSR II) with the FlowJo software (both from BD Biosciences, Franklin Lakes, NJ, USA) as CD29-, CD90-, and CD105-positive and simultaneously CD45-negative cells. MSCs were stained with endoglin/CD105 biotinylated antibody (BAF1320, R&D Systems, Minneapolis, MN, USA), APC anti-mouse/rat CD29 antibody (BioLegend, San Diego, CA, USA), PE-Cy™5 mouse/anti-rat CD45RA antibody (BD Bioscience, Franklin Lakes, NJ, USA), and FITC anti-rat/mouse CD90.1 (Thy-1.1) [HIS51] antibody (eBioscience, San Diego, CA, USA), which were detected with Qdot 605 streptavidin conjugate (Invitrogen, Waltham, MA, USA). Five million MSCs were injected into the ECM skeletons through cannula in SV. After stationary culture for 1 h, the ECM skeletons were connected to the perfusion system, consisting of a peristaltic pump, tubes, and a chamber, with a flow rate of 1.8 mL/min. Dynamic culture was performed for 3 days, after which the ECM skeletons with MSCs were stained with H&E.

### 2.6. Histology

#### 2.6.1. Immunohistochemistry

Tissue samples (*n* = 2 for each perfusion technique) were fixed in buffered 10% formaldehyde (Merck, Darmstadt, Germany) overnight at 4 °C. Then, tissues were processed in an automatic tissue processor (Leica TP1020, Leica Biosystems, Deer Park, TX, USA), followed by embedding in paraffin. Tissues were cut into 4-µm-thick sections, then deparaffinized in xylene and rehydrated in a graded ethanol series. Sections were either stained with hematoxylin and eosin (H&E, DiaPath S.P.A., Martinengo, Italy) or specific antibodies. For immunostaining, the sections were subjected to heat-mediated antigen retrieval with citrate or TRIS buffer (Vector Laboratories Inc., Newark, CA, USA), according to the manufacturer’s recommendation, for 20 min. Endogenous peroxidase was blocked with 3% H_2_O_2_ in methanol for 20 min. To prevent nonspecific binding, samples were pre-incubated with 1% or 10% normal goat serum (Abcam, Cambridge, UK) in Tween–PBS (Merck, Darmstadt, Germany). The sections were then incubated with primary antibodies overnight at 4 °C. The following antibodies and dilutions were used: anti-collagen IV [EPR22911-127] (ab236640, 1:2000; Abcam, Cambridge, UK), recombinant anti-collagen VI [EPR17072] (ab182744, 1:500; Abcam, Cambridge, UK), anti-entactin/NID [EPR22414-125] (ab254325, 1:2000; Abcam, Cambridge, UK), anti-laminin (ab11575, 1:200; Abcam, Cambridge, UK), anti-fibronectin (ab2413, 1:400; Abcam, Cambridge, UK), recombinant anti-insulin [EPR17359] (ab181547, 1:20,000; Abcam, Cambridge, UK), anti-glucagon [K79bB10] (ab10988, 1:10,000; Abcam, Cambridge, UK), recombinant anti-somatostatin 28 [EPR3359(2)] (ab111912, 1:500; Abcam, Cambridge, UK), anti-cytokeratin 7 [EPR17078] (ab181598, 1:500; Abcam), recombinant anti-CD31 [EPR17259] (ab182981, 1:1000; Abcam, Cambridge, UK), anti-vitronectin [ST49-02] (MA5-32157, 1:200; Invitrogen, Waltham, MA, USA), and anti-α-amylase (A8273, 1:1600; Merck, Darmstadt, Germany). The secondary antibody was goat anti-rabbit conjugated to peroxidase (PI-1000, 1:300; Vector Laboratories, Newark, CA, USA), which was applied for 1 h at room temperature (RT). Then, the DAB Substrate Kit (ab64238, 1:100; Abcam, Cambridge, UK) was applied as the chromogen for 4 min at RT. Next, the sections were counterstained with Mayer´s hematoxylin, dehydrated, and mounted with Pertex (Histolab AB, Askim, Sweden).

Tissue samples that contained pancreatic islets labeled with iron nanoparticles were stained with Prussian blue solution (2% hydrochloric acid mixed with 2%potassium ferrocyanide) for 30 min to detect iron deposits.

#### 2.6.2. Immunofluorescence

Tissue samples (*n* = 2 for each perfusion technique) were fixed in buffered 4% formaldehyde (Merck, Darmstadt, Germany) overnight at 4 °C and rinsed with PBS. Next, tissues were submerged overnight in 30% sucrose, embedded in optimal cutting temperature mounting medium (Tissue Teck^®^ O.C.T. compound, Sakura Finetech, Alphen aan den Rijn, The Netherlands), frozen in liquid nitrogen, and stored at −80 °C. After several washes in PBS, 5-μm sections were incubated in blocking solution that contained 5% normal goat serum (Jackson Immunoresearch Laboratories, West Grove, PA, USA) in 0.2% Triton X-100, 0.1 mol/L glycine (both chemicals from Merck, Darmstadt, Germany), and PBS for 1 h at RT, to prevent nonspecific binding. Next, sections were incubated with primary antibodies in a blocking solution for 1 h at RT. The following primary antibodies and dilutions were used: anti-laminin (ab11575, 1:100; Abcam, Cambridge, UK), anti-fibronectin (ab2413, 1:100; Abcam, Cambridge, UK), anti-collagen IV (ab6586, 1:100; Abcam, Cambridge, UK), anti-collagen VI (ab6588, 1:100; Abcam, Cambridge, UK), anti-entactin/NID (ab14511, 1:100; Abcam, Cambridge, UK), and anti-collagen I (R1038, 1:100; OriGene, Rockville, MD, USA). After intensive washing with PBS, sections were incubated with the secondary antibody for 1 h at RT. The secondary antibody was Alexa Fluor Plus 555 goat anti-rabbit immunoglobulin G (A32732, Invitrogen, Waltham, MA, USA), diluted in the blocking solution. Cell nuclei were labeled with 4,6-diamidino-2-phenylindole (DAPI, Merck, Darmstadt, Germany) at a concentration of 5 μg/mL for 5 min at RT. After rinsing with PBS, sections were mounted with antifade solution and examined with a fluorescence microscope (EVOS FL Auto Cell Imaging System, Thermo Fisher Scientific, Waltham, MA, USA).

### 2.7. Isolation of Pancreatic Islets

Pancreatic islets were isolated according to a standard isolation protocol [26]. Briefly, a collagenase solution (1 mg/mL; Merck, Darmstadt, Germany) was injected through the bile duct into the donor pancreas. After a pancreatectomy, digestion was allowed to continue at 37 °C. Pancreatic islets were then separated from the exocrine tissue with a discontinuous Ficoll (Merck, Darmstadt, Germany) gradient. Islets were cultured in CMRL-1066 medium (PAN-Biotech GmbH, Aidenbach, Germany) with 10% fetal bovine serum (FBS), 5% HEPES buffer, 1% penicillin/streptomycin solution, and 1% Glutamax (all chemicals from Merck, Darmstadt, Germany). Islets were stabilized overnight in a humidified incubator at 37 °C and 5% CO_2_ atmosphere. Next, the isolated pancreatic islets were manually handpicked and counted under dissection microscope (Olympus SZH10, Tokyo, Japan) for use in repopulating the ECM skeletons.

The isolated pancreatic islets were labeled with superparamagnetic iron oxide (SPIO) nanoparticles by adding the MRI contrast agent, ferucarbotran (5 µL/mL; Resovist^®^, Schering AG, Berlin, Germany), into the culture medium overnight. This label allowed tracking the islets after transplantation in selected recipients.

### 2.8. Transplantation Experiments

Isolated pancreatic islets in ECM skeletons were transplanted into the greater omentum (*n* = 4) as described in our previous study [23]. Briefly, after a laparotomy the greater omentum was pulled out of the abdominal cavity and spread out on a sterile field. An ECM skeleton was placed on the omentum, and pancreatic islets (resuspended in 0.2 mL Hanks’ balanced salt solution supplemented with 1% FBS) were slowly, manually infused into the ECM skeleton through the SV. Then, the catheter was flushed, removed from the vein, and the vessel was ligated to prevent leakage. The omentum was pulled over the ECM skeleton to enclose it, and the wrapping was secured with a single stitch. The omentum with ECM skeleton was returned into the abdominal cavity. Recipients were monitored with MRI immediately after the transplantation, then every week for 49 days. On days 21 and 49, the grafts were explanted for histological and electron microscopic examinations.

### 2.9. Magnetic Resonance Imaging

All in vivo experiments were acquired on a 4.7 T MR scanner equipped with a dual ^1^H/^19^F surface coil custom-designed and constructed in an MR laboratory. The measurement protocol consisted of a standard fast low angle shot MR sequence with the following parameters: spatial resolution = 254 × 254 µm^2^, slice thickness = 1 mm, scan time (ST) = 8 min 56 s, repetition time (TR) = 130 ms, and echo time (TE) = 3.715 ms, in both axial and coronal slicing orientations. On days 0 and 21, higher spatial resolution imaging (127 × 127 µm^2^) was performed with a longer ST (18 min 14 s). MRI processing and analyses were performed with ImageJ software (https://imagej.nih.gov/ij/, version 1.46r, National Institutes of Health, Bethesda, MD, USA). The parameters examined included the graft volume and the fractional signal loss (FSL), which indicated the presence of iron in the graft. The FSL was calculated as the difference in signal intensities between the ECM skeleton and a reference surrounding tissue, divided by the reference signal intensity, and expressed as a percentage.

### 2.10. Advanced Environmental Scanning Electron Microscopy

A morphological study of the ECM skeletons was realized with an advanced environmental scanning electron microscope (A-ESEM) [27], in combination with the low temperature method (LTM) [28,29]. Briefly, tissue samples were fixed in buffered 4% formaldehyde overnight. Small parts of ECM skeletons and pancreases (up to 5 × 5 mm) were cut for samples, washed 2 times in distilled water, and placed on a cooled specimen holder (Peltier stage). The LTM procedure began with cooling to 0 °C, then the pumping process in the specimen chamber was started as sample cooling continued to −22.5 °C. All experiments were carried out in our customized QUANTA 650 Field Emission Gun SEM (Thermo Fisher Scientific, Waltham, MA, USA), under the following constant operating conditions: 150 Pa water vapor pressure, 10 kV beam accelerating voltage, 45 pA beam current, and 7.5 mm working distance. Sample topography was imaged with an ionization secondary electron detector, equipped with an electrostatic separator. The material contrast of the iron nanoparticles was studied with a scintillation detector to detect the backscattered electrons [30]. Macrographic images were taken of the ECM skeletons, and the whole image was composed by merging micrographs with Maps software (Thermo Fisher Scientific, Waltham, MA, USA).

### 2.11. Statistical Analyses

The results are expressed as the mean ± standard deviation (SD). The Lilliefors test was performed to check the data for a normal distribution. Based on normality, data were evaluated with the Mann–Whitney *U*-test or the *t*-test, as appropriate; *p*-values < 0.05 were considered statistically significant.

## 3. Results

### 3.1. Perfusion Decellularization Efficiency

Donor rat pancreases were perfused via pancreatic duct, portal vein, gastric artery, and splenic vein. Decellularization efficiency was first observed macroscopically. During the decellularization process, the native pancreas continuously changed color from pink-beige to white-translucent (Figure 1a–c). At the end of the decellularization process, after 6 h, acellular skeletons were obtained, with a visible, well-preserved network of vessels-like structures. The integrity of the vascular structures in the decellularized pancreas was tested by applying 400× diluted Patent blue V solution (2.5% *w*/*v*; Guerbet LLC, Villepinte, France). Vascular integrity was confirmed when no leakage into the ECM skeleton was detected (Figure 1d).

The quality of the decellularization process was evaluated by quantifying residual DNA content (Figure 1e). We observed a significant reduction (*p* < 0.001) in double-stranded DNA content in decellularized pancreases (PDP: 9.2 ± 1.9, PDH: 9.4 ± 2.1, PV: 9.6 ± 2.6, GA: 9.5 ± 0.9 and SV: 7.6 ± 1.1 ng/mg) compared to intact control healthy pancreases (143.6 ± 15.4 ng/mg) and diabetic pancreases (124.0 ± 18.2 ng/mg). A significant difference in DNA content was also observed between control healthy and diabetic pancreases (*p* < 0.05).

Decellularization quality was also analyzed in terms of the residual insulin content in insulin-producing beta cells (Figure 1f). The results showed a significant reduction (*p* < 0.001) in insulin content in the decellularized pancreases (PDP: 0.076 ± 0.036, PDH: 0.405 ± 0.499, PV: 0.260 ± 0.156, GA: 0.374 ± 0.397 and SV: 0.121 ± 0.174 ng/mg). On average, the insulin content was reduced to 0.17% of the total insulin measured in control healthy pancreases (142.745 ± 53.7 ng/mg). In contrast, after STZ treatment, diabetic pancreases retained about 10% (14.317 ± 10.0 ng/mg) of the total insulin measured before STZ treatment.

In addition, the acellular composition of the pancreatic scaffolds was confirmed in a histology examination. H&E staining indicated no remnant cells, and DAPI stain showed that no nuclear components were retained after the decellularization process was complete (Figure 2).

### 3.2. Composition of ECM Macromolecules in Decellularized Pancreases

The decellularization process produced skeletons that comprised tissue-specific ECM; however, the use of strong detergents can be disruptive to ECM proteins. Morphological analyses revealed the exact composition of ECM proteins in the pancreatic skeletons (Figure 3). Based on the intensity of the DAB color and red fluorescence, we were able to recognize three levels of the signal: very strong (clearly presented in all parts of the section), strong (presented in most parts of the section), and weak (presented in some parts of the section). 

We found that the ECM skeletons had very strong signal for collagen IV and laminin. Strong signals were also detected for entactin, fibronectin, and collagens VI and I. Collagen IV and VI were particularly abundant in ducts and vessels. Vitronectin elicited the weak signal, indicating nearly zero occurrence in the native pancreas. The proportions of relevant antigens did not differ among the pancreases decellularized in different ways or when they were compared to control native pancreases.

ECM skeletons lacked endothelial cells, based on the absence of CD31 antibody detection. Similarly, ECM skeletons lacked pancreatic-islet hormones (insulin, glucagon, and somatostatin) and the exocrine tissue enzyme, amylase, based on specific antibody detection. The effectiveness of decellularization was also evidenced by the absence of cytokeratin 7, a marker of ductal cells.

DAPI co-staining was used to visualize cell nuclei, and the lack of staining confirmed the efficiency of decellularization (Figure 2). Residual DNA fragments were seen only rarely in ECM skeletons; these were probably not washed out and remained attached to extracellular proteins. In contrast, intact cell nuclei were widely observed in control native pancreases.

### 3.3. Cytocompatibility Testing of the ECM Skeletons

MSCs infused into the ECM skeletons through SV were dynamically cultured for 3 days to assess the cytocompatibility of the prepared ECM skeletons. MSCs were visualized in vessel-like structures of the skeletons, and histological examination did not reveal any signs of the necrosis in cell nuclei (Figure 4). We can therefore assume that ECM skeletons’ microenvironment is sufficiently safe to enable short-term survival of this particularly sensitive type of repopulated cells.

### 3.4. Transplanting ECM Skeletons with Pancreatic Islets into the Omentum

Transplanting pancreatic islets into the omentum was shown to be a promising alternative method that was as effective as the conventional method of transplanting pancreatic islets into the liver [23]. The omentum has a large surface area that is easily accessible for surgery. It enables physiological portal vein drainage of insulin, and it avoids direct contact with the recipients’ blood. However, in rats, the omentum is not sufficiently large to cover an entire pancreas skeleton. Therefore, a new technique was designed for perfusion through the splenic vein (Figure 5a). The pancreatic ECM skeletons that were decellularized with various approaches were compared according to their mean weights (Table 1). Due to differences among rat donors, the weights of the pancreatic skeletons are expressed in mg of wet weight per gram of the donor´s body weight. Significant differences were observed between all the ECM skeletons and the control pancreases (*p* < 0.001), between the ECM skeletons from whole pancreases (PDP, PDH, PV), and the ECM skeletons from different pancreatic segments (SV, *p* < 0.001 and GA, *p* < 0.01).

For the transplantation experiments, we used only ECM skeletons obtained by perfusing the decellularization solutions via the splenic vein (Figure 5a). First, the greater omentum was gently pulled out of the abdominal cavity and carefully spread out. Then, the ECM skeleton was placed on the omentum, and the pancreatic islets were slowly infused into the skeleton through a catheter placed in the SV (Figure 5b).

We first tested the islet infusion into the ECM skeletons in vitro to avoid losing pancreatic islets. We found that applying the islets with a syringe in a volume of 1 mL resulted in losing 11.4 ± 4.6% of the islets. The most effective approach was to apply the islets with a micropipette in a volume of 0.2 mL, followed by flushing as needed. Using that approach, the islet loss was reduced to below 1% (0.7 ± 0.5%). After applying the islets, the SV was ligated to prevent leakage, and the catheter was removed. Finally, the omentum was pulled over the pancreatic skeleton on all sides to enclose the skeleton (Figure 5c). Then, the omentum wrapping around the skeleton was secured with a single stitch. Finally, the omentum with graft was returned back into the abdominal cavity. Based on our practical experience, we found that ECM-skeleton recipients should be of similar or heavier weight than the pancreas donors to ensure sufficient packaging.

### 3.5. Multimodal Imaging of Transplanted Islets in ECM Skeletons

The recipients of transplanted ECM skeletons that contained pancreatic islets labeled with iron nanoparticles were evaluated with ^1^H MRI at several time points (Figure 6a). The initial mean volume of the graft immediately after transplantation was 348.61 mm^3^, and by the first two weeks, its size significantly decreased to a mean of 55.51 mm^3^. This decrease in size was expected, because the fluid contained in the ECM skeleton in the pre-implantation state and during islet transplantation was expected to drain out into the surrounding tissue. Then, the ECM skeleton volume remained unaltered until day 21. On day 35, another decline in size was observed to a mean of 32.88 mm^3^, and after that, the volume remained essentially constant. At the last time point, day 49, it was very difficult to detect implanted ECM skeletons, due to the low specificity of ^1^H MRI. The low volume of the graft made it difficult to distinguish from other structures in the peritoneal area, such as the intestines; therefore, the ^1^H MRI evaluations were discontinued. The FSL calculated at each time point remained stable, at 28.32 ± 2.3%, throughout the entire study, which indicated that the SPIO nanoparticles had not been washed out of the ECM skeleton.

Before transplantation, histological examinations identified iron nanoparticles inside the pancreatic islets and on the islet surfaces. Twenty-one and forty-nine days after ECM skeletons were transplanted into the omentum, the iron nanoparticles were mostly detected in tissues surrounding the pancreatic islets while minor iron deposits were identified also inside pancreatic islets (Figure 6b,c).

Integration of the islets inside the ECM skeletons was verified with advanced environmental scanning electron microscopy (Figure 7). Immediately after placing the pancreatic islets into the skeleton, the islets were, for the most part, densely localized in vessels. At 21 days after transplantation into the omentum, the islets were partially dispersed in the ECM. Similar to the histological findings, we found that the iron nanoparticles were mostly localized outside the pancreatic islets, in contact with the ECM, although some iron deposits were also detected inside the pancreatic islets.

## 4. Discussion

Transplanting islets into the liver has several notorious limitations that could be potentially overcome by placing the islet graft into alternative, extrahepatic locations. Among the alternatives, the greater omentum has shown promise, both experimentally [2,23] and in human pilot trials [24,25]. The advantages of the omentum include its high vascularization, its easy surgical accessibility, and its large surface area, which allows transplantation of larger and less purified or encapsulated islet grafts and the possibility of graft retrieval, when needed [31]. However, the most important advantage is that transplantation into the omentum eliminates the risk of direct contact with the recipient’s blood. On the other hand, previous experimental and clinical results have not shown any advantage of omental implantation over the traditional liver implantation. This finding was probably due to the low oxygen supply [32] and islet migration from the original placement site, despite the use of a biocompatible plasma–thrombin gel to fix the islets within the omental flap and the use of a biocompatible scaffold for engraftment support and revascularization.

Therefore, we reasoned that a decellularized pancreas skeleton might provide a more secure matrix for islet implantation into an omental flap. However, the previously described techniques for preparing pancreatic skeletons resulted in a large size that exceeded the technical feasibility for placing it into the omentum. Consequently, we developed a novel method for decellularizing only the pancreatic tail with an approach through the SV.

Here, we showed that the skeletons prepared with our novel method were of adequate size, and their protein composition, decellularization quality, residual DNA contents, and anatomical integrity were comparable or superior to other decellularization techniques described previously. Although the rat omentum is easily accessible for surgery, it is not sufficiently large to enclose a skeleton of the entire pancreas, prepared with the conventional method. Therefore, in the present study, we implanted only the pancreatic tail skeletons, prepared with SV perfusion, into the omentum. Moreover, we found that, technically, the best repopulation results were achieved when islets were implanted through a micropipette introduced into the SV catheter in a 0.2-mL volume of medium, followed by flushing as needed. This procedure led to only negligible islet leakage from the skeleton.

We confirmed that the islet graft position was stable within the omental flap with repeated MRI monitoring of islets that had been labeled in vitro with iron nanoparticles. Iron nanoparticles were histologically detected inside the islet cells as well as in local macrophages in their vicinity. This finding is in agreement with our previous results showing that the nanoparticles are partly translocated from cytoplasmic vesicles of the endocrine cells into the intercellular space with subsequent phagocytosis by host-derived macrophages [33,34]. The difficult detection of the implanted islets on day 49 on MRI may be due to various reasons. Based on the results of histological examination or electron microscopy, we assume that it was caused rather by sensitivity limits of the ^1^H MRI method than by the absence of the islets in the graft. The visualization of iron nanoparticles (hypointense spots) in peritoneal area may be influenced by artifacts caused by the presence of the air or bowel movements.

In addition, supravital environmental scanning electron microscopy of samples demonstrated that the superficial structures of the islets within the skeletons were well-preserved for up to 21 days after transplantation. Of note, this study was the first to apply this mode of imaging for assessing pancreatic islet (and the decellularized pancreas skeleton) implantation. This technique enables the determination of the shape and positions of objects in their natural environment, without creating artifacts caused by the extensive sample preparation typically required for conventional scanning electron microscopy [35].

Napierala et al. [11] described a successful method for whole rat pancreas decellularization. They compared arterial, PV, and PDP perfusion approaches, and in agreement with our findings, they showed that the choice of decellularization technique had only a minor effect on decellularization efficiency. On the other hand, repopulation approaches showed variable effects. When intact islets were injected into the vascular system, they tended to become trapped inside the vessels, and when islets were injected into the duct, they tended to leak into the pancreatic parenchymal space. Those findings suggested that the preservation of the macroscopic structure of the decellularized organ was less important than the biochemical properties of the skeleton [14,18]. Indeed, in our model, it may be worth exploring further the potential of injecting the islet graft into the exocrine tract, rather than into the vessels of the skeleton.

The remaining unresolved question is, what is the best way to transplant the repopulated pancreatic matrix? In a preliminary study, Napierala et al. [11] investigated islet function in vitro only, as well as Damodaran and Vermette [14], who also demonstrated in vitro cytocompatibility of the prepared scaffold for 120 days. In contrast, Yu et al. [15] described the short-term functionality of INS-1E cells that had been used to repopulate pancreatic skeletons in rats. They described a tricky, vascular anastomosis between skeleton vessels and host renal vessels. That unique study demonstrated the feasibility and patency of the delicate anastomosis of decellularized graft vessels to the recipient circulation. However, it remains unclear whether that technique would be suitable for transplanting intact, whole islets. Similarly, Guo et al. [17] developed a method for the in vitro endothelialization of decellularized rat skeletons with a bioreactor. They subsequently implanted the prepared skeleton into the dorsal subcutaneous space in mice. Although that study may represent an important step toward engineering a bioartificial pancreas, to date, they have not described any attempts to repopulate those pancreatic matrices with beta cells. It has been estimated that pancreatic islets get as much as 15 % of the total pancreatic blood supply [36]. Efforts to achieve sufficient oxygen tension either by direct or stimulated vascularization of beta cell containing cellular constructs seems therefore highly desirable [37].

Out of the typical ECM proteins, we demonstrated abundance of collagen IV, collagen VI, laminin, entactin, and fibronectin. Unfortunately, our analysis did not include the detection of collagen V, which has recently been identified as a critical regulator of islet organogenesis from human pluripotent stem cells and may play a role in beta cell regeneration [38].

In conclusion, this study was designed to develop a suitable model for testing the utility of decellularized pancreatic skeletons as a matrix for seeding an islet graft into the omentum. Pancreatic tail perfusion via the splenic vein provided smaller extracellular matrix, which facilitated transplantation into the omentum. Serial MRIs of iron-oxide-labeled islets demonstrated that the islets maintained their position in the omentum for up to 49 days. Environmental scanning electron microscopy showed that the pancreatic islets retained their characteristic shapes and remained integrated within the skeletons. Histological evaluations and immunofluorescence imaging confirmed their viability and sustained insulin production in non diabetic syngeneic recipients. Further research will show whether the use of decellularized skeletons for islet transplantation will provide superior results in treating experimental diabetes, compared to direct transplantation into the liver or plasma–thrombin gel-supported islet transplantation into the omentum. The proposed model represents also a platform to test additional modifications such as the intravascular versus intraductal islet seeding, the static versus dynamic methods for recellularization, use of supportive mesenchymal stem cells and endothelial cells, and the administration of islet cell aggregates rather then native pancreatic islets [39].

## Figures and Tables

**Figure 1 jfb-13-00171-f001:**
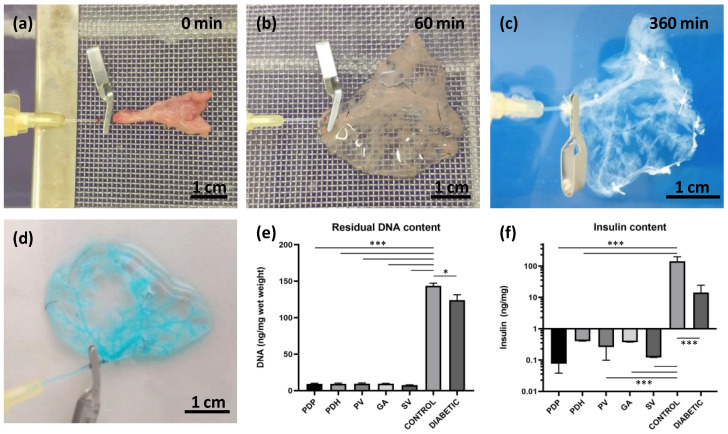
Pancreatic decellularization procedure and assessments. (**a**–**c**) Representative images illustrate the gradual change in color during the perfusion decellularization of the pancreatic tail, via splenic vein (**a**) at the beginning, (**b**) after 60 min, and (**c**) at the end of the decellularization process (360 min); (**d**) the intact vasculature in the decellularized pancreases was evaluated by infusing a patent blue solution, and detecting any leakage; (**e**) DNA quantification; decellularized pancreases contained significantly lower amounts of double-stranded DNA compared to intact healthy control and diabetic pancreases; (**f**) insulin quantification; decellularized and diabetic pancreases contained significantly lower insulin contents compared to control healthy pancreases; perfusion approaches: PDP—pancreatic duct via papilla, PDH—pancreatic duct from hepatic side, PV—portal vein, GA—gastric artery, SV—splenic vein; * *p* < 0.05 and *** *p* < 0.001.

**Figure 2 jfb-13-00171-f002:**
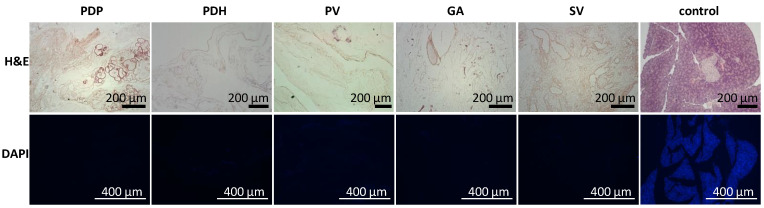
Representative images of the morphology of decellularized pancreases. (Upper row) H&E-stained sections show the lack of remnant cells; (lower row) DAPI stained sections show the absence of nuclear components in pancreatic skeletons perfused with different approaches: PDP—pancreatic duct through papilla, PDH—pancreatic duct from hepatic side, PV—portal vein, GA—gastric artery, SV—splenic vein. H&E—hematoxylin and eosin, DAPI—diamidino phenylindole.

**Figure 3 jfb-13-00171-f003:**
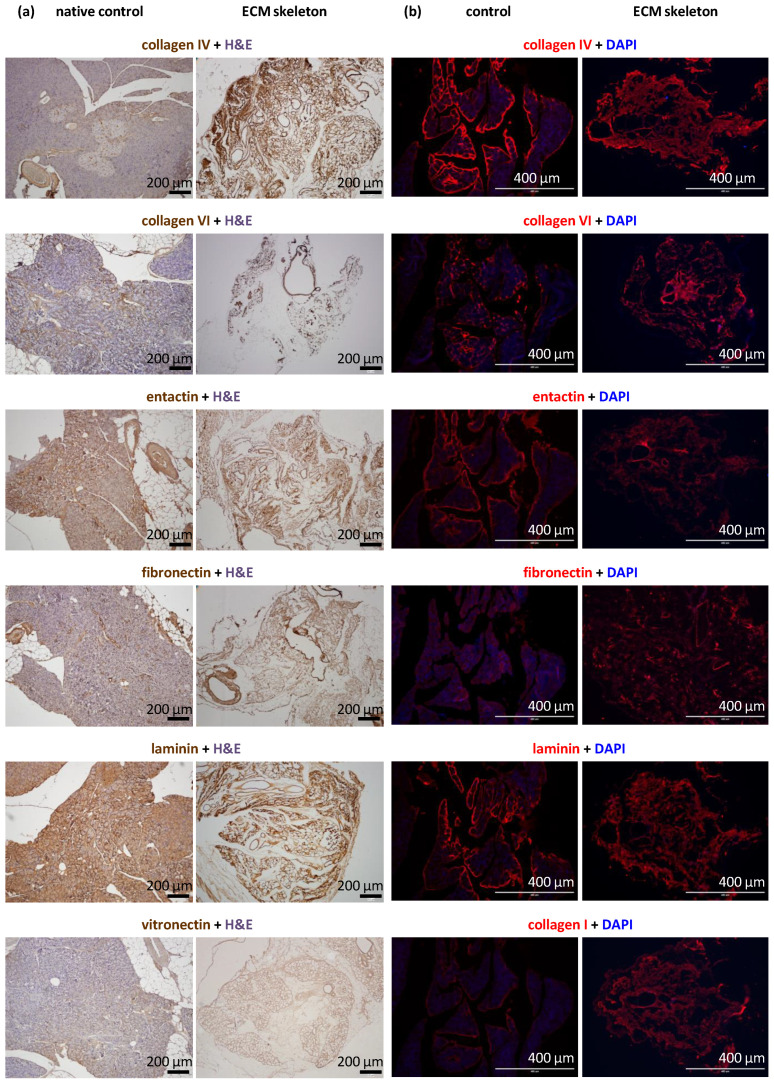
Representative images of the structural proteins in native and decellularized pancreases perfused via the splenic vein. (**a**) Immunohistological images show the identification of selected proteins, including vitronectin, in the control pancreas (left) and the pancreatic skeleton (right), stained with specific antibodies and visualized with the DAB detection system; (**b**) immunofluorescence images show the identification of selected proteins, including collagen I, in the control pancreas (left) and the pancreatic skeleton (right), stained with specific fluorescent antibodies (red); sections were counterstained with DAPI (blue) for detection cell nuclei.

**Figure 4 jfb-13-00171-f004:**
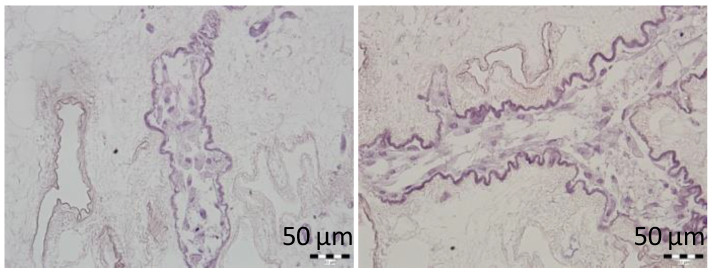
Representative images of the MSCs cultured in decellularized pancreases. H&E-stained sections show the cell survival after 3-day culture in vessel-like structures of the pancreatic skeletons.

**Figure 5 jfb-13-00171-f005:**
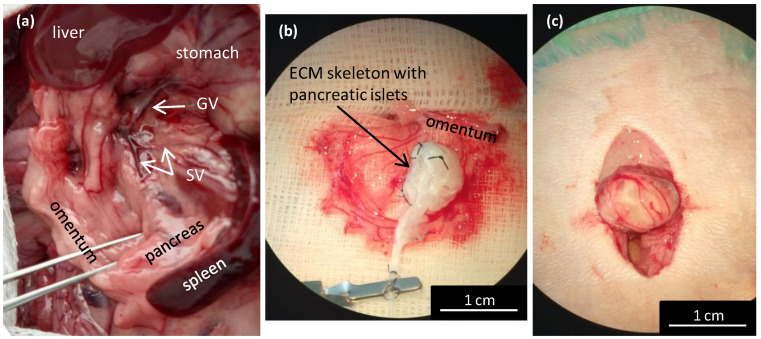
Wrapping the repopulated pancreatic skeleton in omentum for transplantation. (**a**) Illustrative image of the anatomic location of the splenic veins for the decellularization perfusion; GV—gastric vein, SV—splenic vein; (**b**) the ECM skeleton is placed on the extruded omentum; ECM—extracellular matrix; (**c**) the ECM skeleton is packaged in the omentum, and the wrapping is secured with a single stitch.

**Figure 6 jfb-13-00171-f006:**
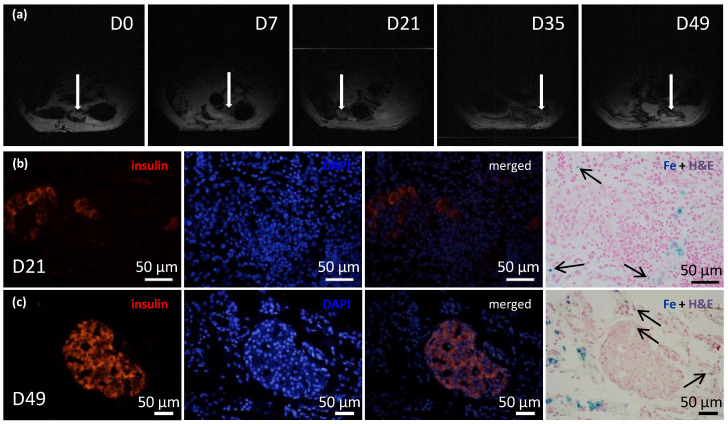
Time course of iron-oxide-labeled pancreatic islets within transplanted ECM skeletons. (**a**) Representative MR images show the ECM skeleton (arrows), which contained pancreatic islets, transplanted into the greater omentum in a rat model over a period of 7 weeks; (**b**) representative fluorescent images of iron-oxide-labeled islets in ECM skeletons, 21 days after transplantation into the omentum; (*left*) insulin-positive cells (red) in the graft; (*middle left*) DAPI counterstaining (blue) indicates cell nuclei; (*middle right*) merged image shows insulin in cells with intact nuclei; (*right*) histological examination of iron deposits (blue stain and arrows), mostly detected in tissue surrounding the pancreatic islets; (**c**) representative fluorescent images of iron-oxide-labeled islets in ECM skeletons, 49 days after transplantation into the omentum; (*left*) insulin-positive cells (red) in the graft; (*middle left*) DAPI counterstaining (blue) indicates cell nuclei; (*middle right*) merged image shows insulin in cells with intact nuclei; (*right*) histological examination of iron deposits (blue stain and arrows), mostly detected in tissue surrounding the pancreatic islets.

**Figure 7 jfb-13-00171-f007:**
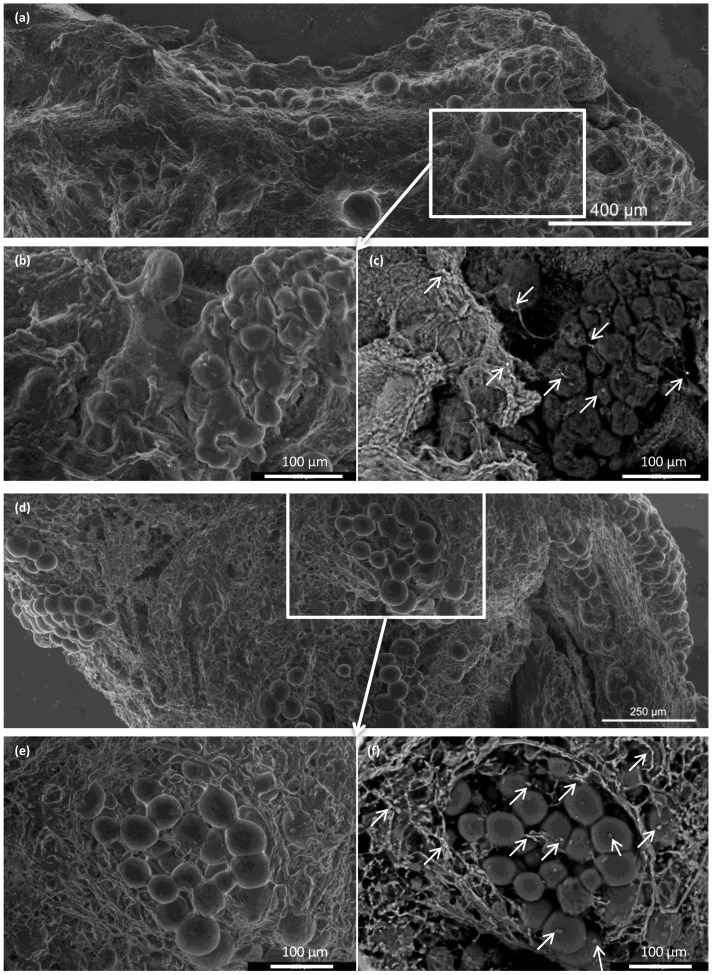
Representative advanced environmental scanning electron microscopy images of pancreatic islets before and after transplantation. (**a**–**c**) Pancreatic islets implanted into the ECM skeleton before transplantation; (**a**) large-field surface morphology of the ECM skeleton shows several clusters of pancreatic islets; (**b**) detail shows the high plasticity of islet cells, (**c**) visualization of iron nanoparticles, areas with high mean atomic numbers appear brighter (arrows); (**d**–**f**) pancreatic islets in the ECM skeleton 21 days after transplantation into the omentum; (**d**) large-field surface morphology shows the explanted graft, (**e**) detail shows the pancreatic islets are partially dispersed in the extracellular matrix of the graft; (**f**) iron nanoparticles in the graft are bright (arrows), and they were mostly localized in the extracellular matrix surrounding the islets, but they were also detected inside the islets.

**Table 1 jfb-13-00171-t001:** The weights of pancreatic ECM skeletons according to the perfusion approach.

Decellularized Pancreases (mg/g b.w.)					Controls (mg/g b.w.)	
PDP	PDH	PV	GA	SV	Healthy	Diabetic
1.16 ± 0.07	1.14 ± 0.10	1.17 ± 0.09	0.54 ± 0.16	0.43 ± 0.09	3.54 ± 0.56	3.24 ± 0.41

ECM—extracellular matrix, b.w.—body weight, PDP—pancreatic duct through papilla, PDH—pancreatic duct from hepatic side, PV—portal vein, GA—gastric artery, SV—splenic vein.

## Data Availability

The data presented in this study are available on request from the corresponding author.

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
