# Peer review of "Decellularized Pancreatic Tail as Matrix for Pancreatic Islet Transplantation into the Greater Omentum in Rats"

_jfb, 2022, doi:10.3390/jfb13040171_

Round 1

Reviewer 1 Report

Berkova et al. present a well-written and clear manuscript. They tested different decellularization approaches and identified that decellularization through the splenic vein was the superior approach. They characterized the maintenance of ECM components such as COL IV, COL VI, entactin, fibronectin, laminin, and COL I. They implanted the pancreatic scaffolds seeded with islet into the omentum for up to 49 days but showed histology on day 21. Although the authors clearly characterize the ECM after decellularization, they do not discuss the toxicity of the decellularized agents that may affect cellular behavior and implantation. 

The authors use a combination of 1% Triton X-100 followed by perfusion with 0.5% sodium dodecyl sulphate (SDS) and then perfusion with 1% Triton X-100. The use of SDS is known to damage the composition of the ECM and is difficult to remove fully. Thus, this leads to adverse cellular behavior, such as cell death. To better demonstrate that this is not the case, a test with seeding of cells such as fibroblasts and assessing their attachment and survival over time may help clarify this isn’t the case. 

Secondly, the images of the iron oxide labeled islets after 21 days in vivo show that the islet shown does not express the iron oxide labeling, which is only present in the surrounding cells. The authors state that “these results are consistent with previously published studies dealing with pancreatic islets labeling and tracking [29,30]” however don’t provide further explanation. It is unclear if the cells migrated out or macrophages consumed the cells, and further clarifications would help.

Thirdly the authors state that they are unable to visualize through MR imaging of the scaffold in vivo after day 21 and do not provide further analysis of the samples. It would be interesting to see the whole scaffold on days 21 and 49 to understand if it is degraded in vivo or has a fibrotic encapsulation. The authors also do not show if the insulin produced by the implanted islets in the scaffold physiologically helps the animal. It will be critical to see if the implantation method allows for the functional physiological secretion of insulin. Would the authors be able to implant the scaffolds in diabetic mice, check if this decreases their glucose levels over time or cause acute damage, and see if the implant can prevent the increase of glucose in the bloodstream? 

Although the paper is well-written and provides valuable insight into the decellularized methods, the authors need additional experiments to better explain how this decellularization and seeding method is beneficial for implantation. The above suggestions can help better represent the work. 

Reviewer 2 Report

The authors of this manuscript analyze several techniques and perfusion routes for the transpantation of pancreatic islets using decellularized scaffolds. The manuscript provides some new data and could be published after considering the following issues:

1. Introduction section: there are several methods of decellularization that each one affects the ECM in the final decelullarized scaffolds. Those techniques should be at least brifely discribed in the introduction and mention the reasons for choosing a technique in this work.

2. Lines 179-184, in the M&M section the details of materials and chemicals used are not well specified. Some times the manufacturer's name is missing, while some times there are no reference numbers.

3. Statistical analysis. The size of the samples/number of replicates per condition for statistical analyses performed in this study are never mentioned in the relevant M&M section or anywhere else in the results, figure legends etc.

4. The statistical significance is not presented on the graphs (only mentioned in the legends or text).

5. Lines 325-329. Herein, there are results described concerning staining against CD31 and cytokeratin 7 that I was not able to find in any figure.

6. About the evaluation of the ECM components by histochemistry there is no description of the evaluation method of the images. To my opinio, since it is generally accepted that the ECM quality is affected after decellularization it is important to describe with more precision the method used for this evaluation or even perform some kind of staining quantifications.

7. Could you specify what is the trichrom staining (is it Masson's) and what is the data extracted from this staining in this particular study?

Reviewer 3 Report

Reviewer Comments:

This is an interesting article “Decellularized pancreatic tail as a matrix for pancreatic islet transplantation into the greater omentum in rats”.  The results presented in the manuscript are in a logical sequence with appropriate statement. This manuscript could be considered after minor revision.

1.       Some references with similar topics and results should be discussed in the context of this manuscript, not only cited in the “introduction” part.

For example, the references 10,15&16 in the manuscript:

10. Goh SK, Bertera S, Olsen P, et al. Perfusion-decellularized pancreas as a natural 3D scaffold for pancreatic tissue and whole organ engineering. Biomaterials 2013; 34: 6760-6772. DOI: 10.1016/j.biomaterials.2013.05.066.

15. Damodaran RG and Vermette P. Decellularized pancreas as a native extracellular matrix scaffold for pancreatic islet seeding and culture. J Tissue Eng Regen M 2018; 12: 1230-1237. DOI: 10.1002/term.2655.

16. Yu HJ, Chen YZ, Kong HR, et al. The rat pancreatic body tail as a source of a novel extracellular matrix scaffold for endocrine pancreas bioengineering. J Biol Eng 2018; 12. DOI: ARTN 610.1186/s13036-018-0096-5.

Other articles with related results should be cited and discussed in the context.

(1) Coronel MM, Stabler CL. Engineering a local microenvironment for pancreatic islet replacement. Curr Opin Biotechnol. 2013 Oct;24(5):900-8. doi: 10.1016/j.copbio.2013.05.004. Epub 2013 Jun 12. PMID: 23769320; PMCID: PMC3783544.

(2) Huanjing Bi, Kaiming Ye, Sha Jin, Proteomic analysis of decellularized pancreatic matrix identifies collagen V as a critical regulator for islet organogenesis from human pluripotent stem cells, Biomaterials, Volume 233, 2020, 119673, ISSN 0142-9612, https://doi.org/10.1016/j.biomaterials.2019.119673.

(3) Hashemi J, Barati G, Bibak B. Decellularized Matrix Bioscaffolds: Implementation of Native Microenvironment in Pancreatic Tissue Engineering. Pancreas. 2021 Aug 1;50(7):942-951. doi: 10.1097/MPA.0000000000001868. PMID: 34643609.

2.      Most of the results(Figures 1~4) showed the results about preparation and identification of decellularized pancreatic tail which were discussed in other articles. Maybe the differences or benefits of this study should be stated and discussed.

3.      The scale bar and notation are not clear in some figures which would be confused.  Maybe more characters such as “ a, b, c, d, e, f, g, h, ……” or “a1, a2, a3, b1, b2, b3..” could be employed to avoid confusion. Some characters are too small to read. The notations in Figure 3 are complicated. It is better to rearrange the results.   

4.      In Figures 5a and 5b, the results with different days should be separated in several isolated side figures to avoid confusion. 

Round 2

Reviewer 1 Report

The authors have sufficiently addressed my concerns. 

Minor suggestion,

Please modify the statement on lines 397 and 398, "therefore we assume negligible or zero toxicity on for this particularly sensitive type of repopulated cells". The authors can not confirm zero toxicity from a cell culture after 3 days where the cells are located within the vasculature. The authors also do not perform staining for cell apoptotic markers, or etc. But rather, the statement should highlight the limited culture of 3 days and state that after 3 days, the cells are visualized in the vasculature. 

Reviewer 2 Report

The authors provided an imroved version of the manuscript.

Only a few minor suggestions are left.

1. Despite the fact that asterisks were added on the graphs to show statistical significance, lines showing pairwise comparisons are missing. 

2. Considering the evaluation of images. To my opinion, even if there was no proper quantification of the stainings there was a methodology for the subjective evaluation that should be described.
